# Prevention of Cirrhosis Complications: Looking for Potential Disease Modifying Agents

**DOI:** 10.3390/jcm10194590

**Published:** 2021-10-05

**Authors:** Giacomo Zaccherini, Manuel Tufoni, Mauro Bernardi, Paolo Caraceni

**Affiliations:** 1Department of Medical and Surgical Sciences, University of Bologna, 40138 Bologna, Italy; giacomo.zaccherini@unibo.it (G.Z.); mauro.bernardi@unibo.it (M.B.); 2IRCCS AOU di Bologna—Policlinico di S. Orsola, 40138 Bologna, Italy; manuel.tufoni@aosp.bo.it; 3Center for Biomedical Applied Research, University of Bologna, 40138 Bologna, Italy

**Keywords:** decompensated cirrhosis, portal hypertension, ascites, non-selective beta-blockers, TIPS, rifaximin, human albumin, statins

## Abstract

The current therapeutic strategies for the management of patients with cirrhosis rely on the prevention or treatment of specific complications. The removal of the causative agents (i.e., viruses or alcohol) prevents decompensation in the vast majority of patients with compensated cirrhosis. In contrast, even when etiological treatment has been effective, a significant proportion of patients with decompensated cirrhosis remains at risk of further disease progression. Therefore, therapies targeting specific key points in the complex pathophysiological cascade of decompensated cirrhosis could represent a new approach for the management of these severely ill patients. Some of the interventions currently employed for treating or preventing specific complications of cirrhosis or used in other diseases (i.e., poorly absorbable oral antibiotics, statins, albumin) have been proposed as potential disease-modifying agents in cirrhosis (DMAC) since clinical studies have shown their capacity of improving survival. Additional multicenter, large randomized clinical trials are awaited to confirm these promising results. Finally, new drugs able to antagonize key pathophysiological mechanisms are under pre-clinical development or at the initial stages of clinical assessment.

## 1. Introduction

In the natural history of liver cirrhosis, the onset of a major complication of the disease represents a crucial clinical event. Indeed, the occurrence of ascites, portal hypertensive gastrointestinal bleeding, hepatic encephalopathy, and deep jaundice, alone or in combination, hallmarks the transition from the compensated to the decompensated phase of cirrhosis. About 5–7% of patients with compensated cirrhosis cross this border every year, thus entering a clinical history punctuated by further complications, frequent hospitalizations, worsening of quality of life and shortening of life expectancy [1].

In this context, the current therapeutic strategies aim to treat or control each complication individually [2]. However, even though their efficacy has been proven in randomized clinical trials (RCTs), such approaches do not significantly affect the course of the disease, thus making it difficult to bring out a clear survival benefit for patients [3,4]. Treatments proven to prevent the progression of the disease are not currently available. However, some potential candidates are under investigation and subject to debate in the scientific community [5,6].

The only interventions showing a clear influence on the natural history of cirrhosis so far are etiological treatments targeted to the elimination of the cause of the disease (i.e., antivirals for hepatitis B and C or prolonged abstinence for alcohol use disorders). In the decompensated stage, even though a slowdown of the disease progression or even a reversion to the compensated state can occur, progression can be seen in up to one-third of cases [4,7,8]. Thus, alternative approaches are warranted.

Recent investigations contributed to uncover hitherto unknown pathophysiological mechanisms of decompensated cirrhosis [9]. It is now clear that the systemic spread of bacterial products from the gut due to abnormal translocation and substances from the liver where inflammation and cell death occur activates immune cells. The consequent release of pro-inflammatory cytokines and chemokines gives rise to sustained systemic inflammation and oxidative stress. These events lead to cardiocirculatory dysfunction responsible for reducing effective volemia and diffuse microvascular damage. These mechanisms provide the pathophysiological substrate for multi-organ dysfunction and, ultimately failure [10].

From this knowledge, novel therapeutic perspectives emerged relying on mechanistic approaches. Their goal is to counteract key pathophysiological events: portal hypertension, bacterial translocation, circulatory dysfunction, systemic inflammation, oxidative stress, and immunological dysfunction. This kind of mechanism is exploited by disease-modifying agents (DMA) who have proven to exert beneficial effects on the course of a disease. These interventions are well established in several fields of medicine, especially in rheumatologic and neurologic disorders. Whether this approach is transferable to cirrhosis (DMAC) has recently emerged as the subject of debate [6]. In any case, some basic principles are unavoidable. Indeed, once a strong rationale for the use of a potential DMAC is established and its efficacy at the pre-clinical level achieved, solid evidence of clinical benefit has to be pursued, preferably through multicenter phase III randomized clinical trials (RCTs) having patient survival as the primary endpoint [11,12]. Given the high clinical heterogeneity of decompensated cirrhosis and its rapidly evolving clinical course [1,13,14], two other characteristics should be clearly identified: the target population receiving the highest benefit from the treatment, and temporary or permanent “stopping rules” in case of loss of efficacy or potential harm.

This review will critically analyze the available evidence supporting that interventions able to counteract key pathophysiological mechanisms can play a role as DMAC in decompensated cirrhosis.

## 2. Etiological Treatments

Whenever possible, removing or neutralizing the cause of the persisting liver insult is the ideal goal to be attempted in all patients with chronic liver disease. Indeed, successful etiological treatments can halt or at least slow down the progression of cirrhosis [5,15].

This is particularly true in the compensated stage of the disease, when the removal of the causative agent generally prevents decompensation, reduces the risk of hepatocellular carcinoma (HCC) and prolongs survival. Thus, etiological treatments undoubtedly represent the first choice in this setting [15]. Etiological treatments remain of primary importance in decompensated cirrhosis, but the extent of their effects is variable. Indeed, the successful elimination of the causative agent can temper the turbulent course of decompensated cirrhosis, or even induce the regression of the disease to a significant extent. However, a portion of patients do not obtain a relevant clinical improvement, remaining at risk of further decompensation, poor quality of life, and reduced life expectancy [5,7,16].

Patients with HBV-related cirrhosis should be treated with specific anti-viral drugs as evidence shows that clinical improvement occurs in most cases [16,17,18]. Once the decompensated stage has been reached, improvement in the severity of cirrhosis and survival rate occurs in about half to two-thirds of cases [19]. There is solid evidence that clinical improvement generally occurs once viral clearance is obtained in patients with HCV-related cirrhosis [20,21]. However, long-term effects on the occurrence of complications and survival remain less defined. First, not all patients benefit to the same extent. Patients with severe portal hypertension and advanced disease, as defined by a model for end-stage liver disease (MELD) score exceeding a certain threshold (between 15 and 20), seem to be less susceptible to sustained clinical improvement [7,21,22]. Second, most patients with decompensated cirrhosis followed for a median period of 4 years after viral clearance did not significantly improve their liver function and MELD score remaining at risk for severe complications, death and need of liver transplantation [8]. Another important issue is related to patients waitlisted for liver transplantation that achieve a modest reduction in the MELD score after HCV viral clearance without improving their clinical conditions to a substantial extent. As a result, they remain at risk of complications and death, yet their priority on the waiting list is negatively affected. Thus, an appropriate evaluation of the timing of antiviral therapy is needed [23,24].

Alcohol withdrawal and abstinence from alcohol are crucial to improving the clinical course and outcomes in patients with chronic alcoholic liver disease, regardless of their disease state [2]. In patients with compensated cirrhosis, such an approach can prevent disease progression and decompensation [15], while in decompensated patients it can at least prevent further deterioration and stabilize the clinical course in most cases [25,26].

No clearly effective and approved drugs are available for patients with non-alcoholic fatty liver disease (NAFLD), so the treatment cornerstone relies on correcting or attenuating the cofactors of the metabolic syndrome, such as optimizing glycemic control in diabetic patients and or lowering body weight in case of obesity. However, great caution is needed in recommending unsupervised weight loss in patients with cirrhosis, since sarcopenia is a highly prevalent and impactful comorbidity [27,28] that can be worsened by incautious weight loss. However, a monitored weight loss associated with lifestyle interventions (like tailored dietary counseling and adapted physical activity) can obtain improvement in portal hypertension and general conditions [29], thus rendering patients more suitable for liver transplantation.

Finally, sporadic data are only available on the effects of etiological treatments in patients with cirrhosis of less frequent etiologies. Indeed, the beneficial effects of removing causative factors in these settings have yet to be demonstrated. The sole exception is autoimmune cirrhosis, whose response to immunosuppressive therapy is followed by an improvement in long-term outcomes [30]. However, an appropriate balance between effective immune suppression and risk for infection can be difficult to achieve in this context.

Summarizing the available evidence, etiological treatments are often effective in slowing down the course of cirrhosis and are of primary importance in the management of patients. However, they present limitations in the decompensated stage of the disease [2,5]. Indeed, once cirrhosis has reached a self-perpetuating “point of no return”, even the removal of the causative agent cannot arrest the disease progression so that the patient long-term prognosis is not influenced. Therefore, the effective management of patients with advanced cirrhosis also requires the adoption of strategies able to counteract the key pathophysiological mechanisms.

## 3. Pathophysiological Treatments

Besides new approaches still under development, interventions directed at counteracting pathophysiological mechanisms include treatments currently used to manage specific complications of cirrhosis. However, to employ them as DMAC, they should be handled in the context of novel strategies. This section will review the available treatments, both already tested and under investigation, able to antagonize one or more key events in the complex pathophysiological network of decompensated cirrhosis (Figure 1).

### 3.1. Non-Selective β-Blockers

Non-selective β-blockers (NSBBs) were introduced in the management of patients with cirrhosis about 40 years ago when their lowering effect on portal pressure was demonstrated. At present, NSBBs still are the only drugs recommended for long-term treatment of portal hypertension [2,31,32].

Propranolol was the first NSBB employed. Its effect on portal pressure is mainly related to the reduction in cardiac output and the unopposed α-adrenergic tone in the splanchnic arterial bed. As a result, portal pressure declines because of a reduced splanchnic blood inflow [33]. About 20 years ago, carvedilol was also introduced in the management of portal hypertension [34]. This NSBB is provided with intrinsic anti-α1-adrenergic activity, contributing to lower portal pressure by decreasing intrahepatic vascular resistance. This characteristic, however, may induce a greater reduction in systemic blood pressure [35]. NSBBs also exert non-hemodynamic effects, as they decrease intestinal permeability, bacterial translocation, and systemic inflammation [36]. Whether these effects are relevant in mediating their effect in humans is still unclear.

The RCTs that evaluated the effect of NSBBs in cirrhosis had the prevention of the first or recurrent variceal bleeding as primary endpoints. Even though most of these trials excluded patients in the advanced stages of cirrhosis, several also included patients with decompensated disease. Interestingly, metanalyses showed that bleeding prevention was more pronounced in decompensated than in compensated patients [37,38].

Unfortunately, data to ascertain if NSBBs administration to patients with decompensated cirrhosis prevents other complications beyond variceal bleeding are insufficient. A first systematic search did not find sufficient evidence supporting a NSBBs effect on other complications of cirrhosis. This negative result was possibly due to the underreporting of non-bleeding complications [39]. Indeed, a more recent meta-analysis pooling data from 15 studies showed that patients lowering portal pressure gradient below 12 mmHg or more than 20% from baseline exhibited lower bleeding rates, lower incidence of complications, and a greater improvement in survival than non-responders [40]. This suggests that NSBBs may substantially modify the course of the disease.

Two recent clinical trials reinforced this concept. The PREDESCI trial showed that NSBBs administration to patients with compensated cirrhosis delays decompensation, mainly preventing ascites formation [41]. Another study [42] assessed the effects of carvedilol, carefully titrated based on blood pressure and heart rate, in patients with acute-on-chronic liver failure diagnosed according to the definition of the Asian-Pacific Association for the Study of the Liver [43]. In this placebo-controlled trial, carvedilol reduced the incidence of bacterial infections and acute kidney injury and lowered mortality.

Soon after the first reports on the use of propranolol in patients with cirrhosis, a word of caution rose on its use in patients with ascites as they may risk for developing hepatorenal syndrome (HRS) [44]. Safety signals on either survival or kidney function did not emerge from the numerous RCTs comparing NSBBs with placebo in patients with cirrhosis. However, most studies did not enroll patients with refractory ascites. Further emphasis on this matter derived from observational studies reporting worse outcomes in patients with refractory ascites [45] or spontaneous bacterial peritonitis (SBP) [46]. The concept of a “therapeutic window” beyond which NSBBs may become detrimental was proposed [47], and several subsequent studies were dedicated to this matter. Unfortunately, conflicting results were provided [48].

Different effects of NSBBs on intrarenal hemodynamics of patients with responsive or refractory ascites may explain their potential adverse effects in the latter. Indeed, as opposed to patients with diuretic responsive ascites, NSBBs lower renal perfusion pressure below the critical threshold for autoregulation (65 mmHg) in 55% of the patients with refractory ascites. This phenomenon is likely due to an excessive reduction of cardiac output due to systolic dysfunction [49]. Thus, the question arises as to how identifying patients who may receive harm from NSBBs treatment. Possibly, arterial pressure is the simplest biomarker to establish when the therapeutic window for NSBBs closes. A prospective observational study reported that mean arterial pressure levels <65 mmHg defines the threshold below which NSBBs do not improve transplant-free survival of patients with ascites, particularly in those with SBP and ACLF [50]. Similar results were observed using the threshold of 90 mmHg of systolic blood pressure, which is recommended in current guidelines as a threshold for dose reduction or discontinuation of NSBBs [2,31,32].

In summary, the current indication for NSBBs administration to patients with cirrhosis remains the prevention of variceal bleeding or rebleeding. Nonetheless, when an effective lowering in portal pressure is achieved, more general effects can be observed on the incidence of complications and patients’ survival, suggesting the potential role of NSBBs as DMAC. However, besides the prophylactic action on portal hypertensive bleeding or re-bleeding, NSBBs effects vary in different stages of the disease. On the one hand, prevention of decompensation (mainly ascites) can be seen in compensated cirrhosis; on the other, renal dysfunction and HRS can occur in patients with very advanced cirrhosis. Therefore, further studies are eagerly needed to identify target subgroups of patients who may benefit most from long-term NSBBs administration.

### 3.2. Transjugular Intrahepatic Porto-Systemic Shunt

As already reported, portal hypertension plays a crucial causative role in most complications of cirrhosis. Therefore, interventions targeted to decrease portal pressure, such as transjugular intrahepatic porto-systemic shunt (TIPS), can potentially modify the long-term clinical course of patients with cirrhosis [51]. To date, the main clinical settings for the use of TIPS are the management of variceal bleeding and the control of refractory ascites. However, the main contentious points remain the identification of target patients and appropriate timing for TIPS placement.

The role of “pre-emptive” TIPS as a potential DMAC emerged from 3 RCTs in patients with cirrhosis and variceal hemorrhage [52,53,54] (Table 1). They showed that TIPS placement within 24 or 72 h is effective in controlling bleeding, preventing rebleeding, and improving survival in patients at high risk of uncontrolled bleeding and bleeding-related mortality (HVPG > 20 mmHg, Child C [10–13 points] or B with active bleeding). Moreover, 2 recent large multicenter observational studies highlighted that “pre-emptive” TIPS also improves survival in patients with acute variceal bleeding and ACLF [55,56]. In patients with persistent bleeding or severe rebleeding within five days, “rescue/salvage” TIPS is used when other therapeutic alternatives are not available, but, despite a high rate of bleeding control, a clear benefit on survival has not been established [31,32]. Similarly, although there is consensus for TIPS insertion as secondary prophylaxis of bleeding in patients who failed to respond to endoscopic banding plus non-selective beta-blockers (NSBBs), a benefit on survival has not been consistently observed likely because of the high heterogeneity of the patients included in the studies [31,32].

When TIPS insertion is related to the treatment of ascites, the available evidence for its capacity of modifying the course of the disease is less clearly defined. So far, seven RCTs compared the effect of TIPS versus large-volume paracentesis plus albumin, the standard of care for patients with refractory ascites [57,58,59,60,61,62,63] (Table 2). TIPS showed a superior efficacy in controlling ascites in all trials, although only shunts performed with covered stents positively affected survival [63]. A similar effect on survival occurred in patients with less advanced disease [62] or “recurrent/recidivant” ascites not fulfilling criteria for refractory ascites [58,61,63]. Furthermore, the high incidence of adverse events, such as hepatic encephalopathy, liver failure, and cardiac dysfunction, requires great caution in TIPS use in these patients [51], thus strengthening the need for a thoughtful selection of target patients. Promising results for expanding the application of TIPS could come from the use of smaller diameter stents (6–8 mm instead of 10 mm), which showed similar efficacy, but a lower incidence of adverse events, likely by preventing excessive shunting [64,65].

The reduction in portal pressure induced by TIPS insertion is associated with beneficial effects on renal function in patients with cirrhosis and ascites, including an increase in sodium and water excretion and reduced activity of vasoconstrictor and antinatriuretic systems [2,51]. This could represent a valid rationale for the use of TIPS also in patients with HRS. The results of the available trials [66,67,68], although limited by a non-controlled design and small sample sizes, support this therapeutical approach, especially for non-transplantable patients. Furthermore, the practical applicability is usually limited by the concomitant severe degree of liver failure [2].

In summary, by acting against a key pathophysiological mechanism as portal hypertension, TIPS has a great potential to serve as DMAC. However, two factors limit its widespread use to this end. On the one hand, available studies evaluated the effects of TIPS in patients who develop either portal hypertensive bleeding or difficult-to-treat ascites. Its impact in patients with clinically significant portal hypertension but without these complications has never been assessed. On the other, the occurrence of severe TIPS-related complications often prevents its insertion in patients with refractory ascites. The use of a small-diameter covered stent may help in overcoming these limitations.

### 3.3. Poorly Absorbable Antibiotics

Acting on the intestinal microbiota to limit the abnormal translocation of bacteria and bacterial products is another treatment strategy of relevant importance in patients with cirrhosis. The use of antibiotics would appear an obvious choice, whose main limitation derives from the induction of bacterial resistance. Most available studies assessed the effects of non- or poorly-adsorbable antibiotics to limit their activities in the intestinal environment.

#### 3.3.1. Quinolones

The enhanced susceptibility to bacterial infections by patients with cirrhosis has long been known. The high prevalence of episodes sustained by Gram-negative microorganisms of intestinal origin led to recognize the relevance of the abnormal bacterial translocation from the gut [69,70,71].

The proof-of-concept of the efficacy of antibiotic prophylaxis derives from an RCT comparing 12 months of treatment with norfloxacin versus placebo for 12 months to prevent SBP recurrence [72]. Norfloxacin remarkably reduced the recurrences due to Gram-negative bacteria, without side effects. Subsequent studies showed that either norfloxacin or ciprofloxacin significantly prevented SBP occurrence in high-risk patients with low protein concentration in ascites, liver failure, and/or gastrointestinal bleeding. The benefit deriving from either norfloxacin or ciprofloxacin in either primary or secondary prophylaxis of SBP was confirmed by most meta-analyses [73,74]. Interestingly, norfloxacin reduced circulating bacterial products associated with improved systemic inflammatory markers and circulatory dysfunction [75].

These results potentially demonstrate a pathophysiological effect beyond SBP prevention. However, a benefit on survival and the occurrence of complications of cirrhosis still needs to be convincingly established. A pivotal study in decompensated patients with a low protein content in ascitic fluid (<1.5 g/dL) and severe cirrhosis (Child–Pugh ≥9 with serum bilirubin ≥3 mg/dL) or impaired renal function (serum creatinine ≥1.2 mg/dL, blood urea nitrogen ≥25 mg/dL or serum sodium ≤130 mEq/L), showed that norfloxacin prophylaxis not only reduced the incidence of SBP but also lessened one-year risk for HRS and improved one-year survival compared with placebo [76]. However, both survival and incidence of HRS were secondary endpoints, and the study sample size was inadequate. More recently, a large multicenter RCT did not demonstrate a survival advantage in Child C patients receiving norfloxacin prophylaxis, although a post-hoc analysis unveiled a reduction in six-month mortality confined to the subgroup with low ascites protein concentration [77]. Finally, concerns about the safety of quinolones being a risk factor for the development of infections due to drug-resistant bacteria have been raised over the years [71], although more recent reports did not confirm these findings [77,78].

In conclusion, several important questions regarding the use of quinolones in decompensated cirrhosis remain unanswered, so that their administration as a DMAC cannot be currently proposed.

#### 3.3.2. Rifaximin

There has been growing interest in the possible efficacy of rifaximin in preventing infections and other complications of portal hypertension in cirrhosis. Rifaximin is a minimally absorbed antibiotic with activity against Gram-negative and Gram-positive bacteria [79]. Moreover, it can improve the gut epithelial layer homeostasis, decrease inflammatory pathways, impair bacterial adhesion to enterocytes, and modulate the gut microbiome, thus positively affecting the entire gut liver-axis [80,81,82].

Currently, the only recognized indication for rifaximin use in patients with cirrhosis is the prevention of recurrent hepatic encephalopathy (secondary prophylaxis) [2], mainly based on the pivotal RCT by Bass et al. in 2010 [83]. However, both observational studies and a few small-scale RCTs showed an association between rifaximin treatment and important clinical outcomes, including better control of ascites [84,85], a reduced incidence of decompensation, hospitalizations, variceal bleeding, SBP, and HRS with a decreased risk of renal replacement therapy [86,87,88,89,90,91,92,93]. Some studies even suggested an improvement in mortality [85,87,89]. However, several metanalyses [94,95,96] highlighted the overall low quality of these studies precluding any generalization on the beneficial impact of rifaximin on the clinical course of decompensated cirrhosis. Therefore, before using rifaximin as a DMAC can be advocated in clinical practice, well-designed RCTs with hard endpoints and adequate sample size are needed.

### 3.4. Statins

The first proposal of statins for treating portal hypertension in cirrhosis dates back to the early 2000s [97] since they reduce intrahepatic vascular resistance by enhancing nitric oxide (NO) production in liver sinusoids. Subsequent investigations in experimental cirrhosis also unveiled anti-inflammatory and hepatoprotective properties [98,99,100,101,102]. Statins exert their effects on liver inflammation and fibrogenesis by upregulating the endothelial Kruppel-like factor 2 (KLF2) [101]. This transcription factor regulates the expression of several vasoprotective genes controlling apoptosis, inflammation, oxidative stress, thrombosis, and vasodilation. Furthermore, it inhibits RhoA/Rho-kinase signaling, which is partly responsible for the contractility of hepatic stellate cells [102].

Several observational studies reported the beneficial effects of statins on various clinical aspects of advanced liver disease. They include lower rates of decompensation, liver cancer, and bacterial infections along with increased survival [103,104]. However, these results require caution due to potential flaws in observational studies [105]. Different RCTs convincingly showed that simvastatin decreases portal pressure [106,107]. Further studies compared the lowering effect on portal pressure induced by combining statins with beta-blockers vs. beta-blockers alone, providing discordant results [108,109].

Only one randomized trial evaluated the effects of statins on the complications of cirrhosis [110]. This trial included patients with recent variceal bleeding and assessed the effects of adding simvastatin 40 mg daily or placebo to the standard therapy to prevent rebleeding (NSBBs and endoscopic variceal ligation). Simvastatin did not significantly influence the primary endpoint, which was a composite of rebleeding and death. Nevertheless, it was associated with a significant survival benefit, mainly related to decreased mortality from bleeding and bacterial infections. Patient subgroup analysis showed that this result was limited to patients belonging to Child-Pugh classes A and B. This apparent contradiction (no prevention of complications but improved survival) suggests that the effect on portal pressure might be less relevant than the non-hemodynamic effects of statins, attenuating the intense inflammatory response triggered by infections or bleeding events [99,100,111,112] that plays a major role in the development of ACLF and mortality [6,113].

In conclusion, although there is a robust rationale suggesting that statins might be beneficial for patients with cirrhosis, the evidence from randomized trials is scarce, limited to a single study with a positive result on a secondary endpoint. Non-hemodynamic effects of statins might be more important than the effects of portal pressure.

The currently ongoing LiverHope Efficacy study, a multicenter double-blind trial comparing the effects of simvastatin plus rifaximin vs. placebo in the prevention of ACLF and mortality (www.liverhope-h2020.eu, accessed on 4 October 2021), should clarify whether statins are effective and safe in advanced cirrhosis. Growing evidence support the role of bacterial translocation and systemic inflammation as key drivers of cirrhosis progression and ACLF development. On this pathophysiological background, the association of an agent effective in preventing bacterial translocation (rifaximin) and a drug with anti-inflammatory properties (simvastatin) could provide a dual effect able to counteract disease progression. It is important to emphasize that although there have been no safety issues in patients with compensated cirrhosis, statin pharmacokinetic is markedly altered in patients with decompensated liver disease [110], with a very high risk of muscle toxicity compared to the general population [110]. Very recent data from the LiverHope double-blind dose-finding safety trial showed that a dose of 20 mg/day was associated with no muscle toxicity [114].

### 3.5. Human Albumin

The well-established recommendations for human albumin (HA) use in patients with decompensated cirrhosis pertain to conditions characterized by an acute worsening of effective volemia. Indeed, one-shot or short-term HA administration is employed to prevent paracentesis-induced circulatory dysfunction (PICD), prevent renal dysfunction induced by SBP, and diagnose and treat HRS in association with vasoconstrictors [2,115].

The oncotic properties of HA make it an optimal candidate to correct or attenuate effective hypovolemia. However, HA exerts several functions not related to its oncotic properties. These pleiotropic functions include binding of damaging molecules, modulating inflammation and immune responses, exerting antioxidant activity, improving cardiac function, and restoring endothelial integrity [116]. Therefore, from a pathophysiological perspective, HA could act as a multitarget agent, potentially modifying the clinical course of decompensated cirrhosis. Such an approach would imply long-term HA administration.

In 2018, two RCTs and one prospective observational study evaluated the efficacy of long-term HA in patients with ascites, opening new perspectives for the treatment of decompensated cirrhosis [117,118,119]. The ANSWER study [117], a multicenter open-label RCT, enrolled patients with uncomplicated grade 2 or 3 ascites. Those patients included in the active arm of the study received 40 g of HA per week for the initial two weeks, then 40 g weekly for a maximal duration of 18 months. The primary endpoint of the study was reached, as patients receiving HA had a significantly better 18-month overall survival, with a 38% reduction in mortality hazard ratio. Moreover, the management of ascites became easier, as the need for paracentesis and the incidence of refractory ascites were reduced by about 50%. HA administration also lowered the incidence rate of the major complications of cirrhosis. As a result, patients receiving albumin needed fewer and shorter liver-related hospitalizations and preserved their quality of life. Similar results were also reported by a single-center, prospective, non-randomized study in patients with refractory ascites [118]. In contrast, the MACHT study [119], a multicenter placebo-controlled RCT performed in patients with ascites waitlisted for liver transplantation, showed no differences in either the probability of developing complications or death between patients treated or not with HA.

These divergent results should not advise against long-term HA use in decompensated patients. Contrariwise, their comparison provides essential information about patients to be treated and, mainly, the dose and schedule of HA administration. The ANSWER and the MACHT studies differed in terms of design, baseline patient characteristics, length of follow-up, and dosage and timing of albumin administration. A consequence of the lower amount of HA given to patients enrolled in the MACHT trial was that serum albumin concentration remained steady throughout the follow-up [119]. In contrast, serum albumin concentration rose by 0.7–0.8 g/L to almost 4 g/dL in the ANSWER study [117]. A post-hoc analysis of the latter trial showed that on-treatment serum albumin concentration at one month predicts the probability of 18-month overall survival, which was greater than 90% in patients whose serum albumin concentration reached levels 4 g/dL [120]. Baseline serum albumin and MELD score value independently predicted the achievement of this threshold. This would imply that patients with severe hypoalbuminema and very high MELD score should receive greater amounts of HA to achieve the best results [120]. Two other pieces of evidence support the importance of steadily increasing serum albumin concentration beyond a certain level. First, in the pilot-PRECIOSA study, only a “high dose” of HA (1.5 g/kg b.w. every week) made serum albumin rise to a concentration close to 4 g/dl, while a lower dose (1 g/kg b.w. every 10 days) failed to normalize serum albumin in most cases. Notably, only patients who received the high HA dose improved their cardiocirculatory function [121]. Second, a serum albumin concentration greater than 4 g/dL is physiologically present in more than 90% of healthy adult individuals [122].

The recently published ATTIRE Trial [123], an open-label, multicenter RCT which included hospitalized patients with acute decompensation of cirrhosis (with or without ACLF) deserves some comments. In this study, albumin was administered for up to 14 days with the goal to maintain a serum albumin level >3.0 g/dL. No differences were observed in the incidence of infections, renal dysfunction, and death between the two study arms. Moreover, some safety concerns have been raised, mainly due to a higher incidence of fluid overload and pulmonary edema in treated patients. Therefore, albumin administration does not appear to modify the short-term course of cirrhosis in severely ill patients admitted to hospital for an acute complication, at least with the dose and schedule of administration chosen in the ATTIRE study. So far, indeed, the efficacy of short-term albumin use has been demonstrated only in patients with SBP or HRS. Acutely ill hospitalized patients, however, represent a completely different clinical setting compared to chronic administration to stable decompensated patients.

In conclusion, there is growing evidence that long-term HA administration can modify the natural history of decompensated cirrhosis, thus acting as a DMAC rather than a specific treatment for ascites and their acute complications. Future studies are warranted to better characterize patient subgroups who could benefit the most from this novel approach. There is also a need to go beyond a fixed dosage and schedule of HA administration, ideally tending to an individualized and patient-tailored approach. Future investigations should prove or challenge this hypothesis and other open issues, including the definition of stopping rules and the cost-effectiveness (in settings other than Italy [117]) of this relatively demanding treatment in terms of logistics and patient adherence.

### 3.6. Granulocyte Colony-Stimulating Factor

Several RCTs assessed the effects of granulocyte colony-stimulating factor (G-CSF), either alone or in combination with bone marrow stem cell transplantation, in patients with acute alcoholic hepatitis (AH) and/or ACLF and in patients with more stable decompensated cirrhosis. Among the pleiotropic effects of GCS-F, the stimulation of liver regeneration and the improvement in immune dysfunction have been proposed as the mechanisms of action [124]. A recent meta-analysis including 7 RCTs [125,126,127,128,129,130,131] in patients with AH and/or ACLF showed a significant benefit of 90-day survival in favor of G-CSF [132]. However, these results were not confirmed by another analysis [133]. Conclusions are also challenging due to the high heterogeneity of the studies and by the different outcomes found in Europe and Asia [132].

Similar results have been reported in RCTs assessing G-CSF in patients with more stable decompensated cirrhosis. Studies from India provided positive results [134,135,136,137]. In contrast, a trial performed in the United Kingdom showed no improvement in MELD score at three months with G-CSF, with or without hematopoietic stem-cell infusion (CD133+) [138]. Moreover, there was an increase in adverse events and sepsis.

RCTs demonstrating the efficacy of G-CSF in European and US cohorts are needed if G-CSF is to be considered a therapeutic option in the future. Aspects to be evaluated include the ideal dose and schedule of administration, the duration of therapy, the use of combinatorial therapies, the identification of the most appropriate target population, a better understanding of the mechanisms of action, including those potentially adverse, and, finally, understanding whether geographical differences affect the response.

## 4. Controversial Areas and Future Perspectives

Based on the data reported above, no treatments are so far available to effectively manage patients with cirrhosis, especially in the decompensated stage of the disease. Besides various etiological treatments, that in patients with compensated cirrhosis can often prevent decompensations and complications, no solid data support the efficacy as a DMAC of any currently available treatment for decompensated patients. Indeed, some interventions given to treat/prevent a specific complication or comorbidities appear to act as a DMAC (i.e., albumin or TIPS) or have the potential to act (i.e., rifaximin, NSBBs, statins, G-CSF) in certain subgroups of patients.

Since the goal of a DMAC is to halt or at least slow down the progression of the disease, or even partially revert decompensation, its efficacy should be tested by RCTs with patients’ survival as a primary endpoint. However, the very large sample size required to reveal statistically significant differences forces researchers to use alternative “surrogate primary endpoints”, such as the incidence of complications and/or ACLF. These latter events, however, would work better as secondary endpoints, which should also include hospitalizations, quality of life, and cost-effectiveness of treatment [11,12]. Unfortunately, up to now, very few published RCTs have addressed the above endpoints. Examples of such trials are the ANSWER and MACTH trials on albumin [117,119] or the NORFLOCIR trial on norfloxacin [77]. In contrast, the low-quality data generated by a long series of observational studies and small-scale RCTs, which even fairly consistently show an improvement in survival and incidence of complications [84,85,86,87,88,89,90,91,92,93], have to date precluded the use of rifaximin beyond the evidence-based indication of the prevention of HE recurrence. As a result, in the future reliable and conclusive data on DMACs can only derive from multicenter, possibly international, well-designed and adequately powered RCTs, thus highlighting the importance of scientific consortia for promoting and coordinating these research projects.

Moreover, due to the complexity and heterogeneity of patients included in the definition of decompensated cirrhosis, some interventions could act as DMAC only in well-defined subgroups of patients. An example is given by albumin, which has been found effective in the ANSWER trial, if administered long-term in patients with stable decompensated cirrhosis and persistent grade 2 or 3 uncomplicated ascites [117], but not in the ATTIRE study, which enrolled severe and acutely decompensated patients admitted to hospital, with or without ACLF [123]. In addition to the target population, other open issues for any intervention include transferability to daily clinical practice, the definition of dosage and schedule of administration, factors guiding treatment, temporary or permanent stopping rules, the use of combinatorial approaches, cost-effectiveness for healthcare systems, and access to treatment worldwide. Hopefully, some ongoing RCTs will clarify whether the course of decompensated cirrhosis could be positively impacted in well-defined groups of patients by drugs or interventions that are currently used for more limited indications (NCT04072601, NCT03780673, NCT02401490, NCT03451292).

The increasing understanding of the pathophysiological mechanisms underlying decompensated cirrhosis and leading to hepatic and extra-hepatic organ failure could potentially provide new interventions, drugs, and biological substances. Novel candidate DMACs should be able to target gut microbiota or key mechanisms in the pathogenetic network of gut-liver axis, systemic inflammation, and immune dysfunction. One such approach, which is already under development, aims to antagonize a crucial upstream step of the inflammatory cascade by TAK-242, an inhibitor of toll-like receptor 4 (TLR4) [139], which binds the circulating pathogen-associated molecular patterns (PAMPs), such as lipopolysaccharides (LPS) and gram-negative endotoxins [140], as well as damage-associated molecular patterns (DAMPs), such as cleaved nucleosomes, histones, and high-mobility group box 1 proteins (HMGB1) [141].

In the context of acutely decompensated disease with an abnormal burst of systemic inflammation, interesting perspectives could derive from studies assessing albumin-based extracorporeal liver assist devices. The molecular adsorbent recirculating system (MARS) and Prometheus provided proof of concept that such a strategy could be successful but did not show improvement in survival [142,143,144]. A novel device, DIALIVE, has been developed to remove and replace the damaged albumin while also removing DAMPs and PAMPs [145]. A multicenter RCT (NCT03065699) aiming to assess the safety and performance of DIALIVE in patients with ACLF has been recently completed and positive results have been publicly announced. Moreover, as an extension of this concept, another phase III, multicenter, RCT on plasma exchange in patients with ACLF (NCT03702920) is currently ongoing. Their results are eagerly needed to improve the management of these severe acute patients.

The identification of new DMACs will be also aided by innovative research technologies and approaches, such as high throughput -omics techniques and systems medicine analysis. When applied to large cohorts of patients with detailed clinical data, treatment history and outcome as well as biological samples, these approaches will support the identification or development of new DMACs and biomarkers to predict patient prognosis and response to therapies, such as the DECISION project (www.decision-for-liver.eu, accessed on 4 October 2021) is currently pursuing in the perspective of personalized medicine.

The next decade will reveal whether patients with cirrhosis, especially in its decompensated stage, could benefit from treatments that globally manage their disease by reducing the occurrence of complications and ACLF, preventing hospitalization, and ultimately improving survival and quality of life, as already occurs for other impactful diseases.

## Figures and Tables

**Figure 1 jcm-10-04590-f001:**
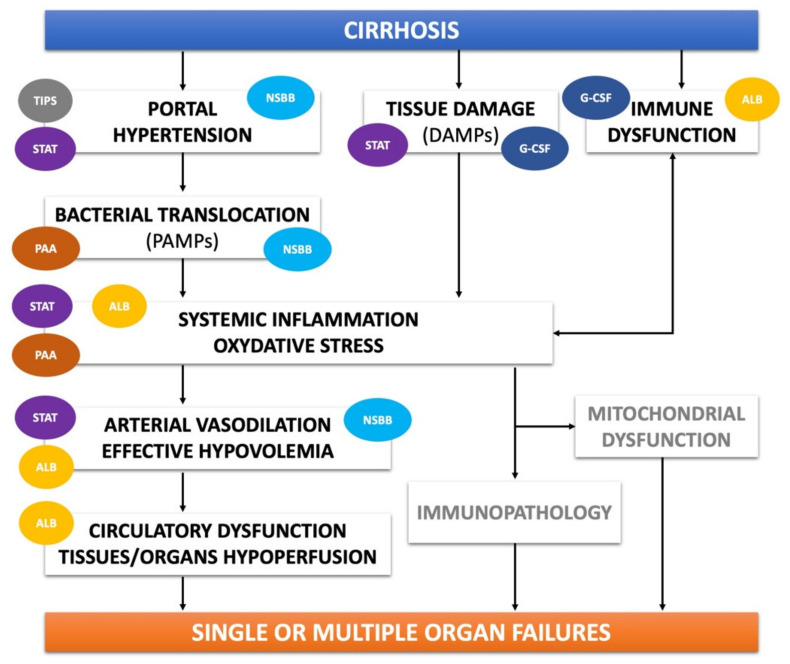
Targets of action of available treatments antagonizing key pathophysiological events in decompensated cirrhosis. See text for details. TIPS: trans-jugular intra-hepatic porto-systemic shunt; STAT: statins; NSBB: non-selective beta-blockers; G-CSF: granulocyte colony stimulating factors; PAA: poorly absorbable antibiotics.

**Table 1 jcm-10-04590-t001:** RCTs assessing pre-emptive TIPS vs. endoscopic treatment in acute variceal bleeding.

Reference	Study Population (Randomized Patients)	Exclusion Criteria	Survival-Related Endpoints of the Study	Effect on Survival
Monescillo A., et al. (Hepatology, 2004)	52 patients with cirrhosis admitted for AVB and HVPG ≥20 mmHg	Age <18/>75 years HCC PVT Previous TIPS HIV infection Chronic heart or renal failure	Primary endpoint: Prospective assessment of treatment failure as well as short- and long-term survival	Mortality reduced by TIPS: In hospital: 11% vs. 31%, *p* = 0.02; ARR 20% 1-year: 38% vs. 65%, *p* = 0.01; ARR 27% Bleeding-related: 19% vs. 38%, *p* < 0.05
Garcia-Pagan JC, et al. (N. Engl. J. Med., 2010)	63 patients with cirrhosis admitted for AVB (CTP B with active bleeding at endoscopyor CTP C ≤13 points)	Age <18/>75 years HCC outside Milan criteria Occlusive PVT Previous TIPS Failure of NSBB plus EVL, Bleeding from GV/ectopic varices Creatinine >3 mg/dL Chronic heart failure	Secondary endpoint: 6-weeks and 1-year mortality	Mortality reduced by TIPS: 6-week: 3% vs. 33%; ARR 30% 1-year: 14% vs. 39%, ARR 25% (*p* = 0.001)
Lv Y., et al. (Lancet Gastroenterol. Hepatol., 2019)	132 patients with cirrhosis admitted for AVB (CTP B patients with and without active bleeding at endoscopy or CTP C ≤13 points)	As above + Recurrent HE (without precipitating factors)	Primary endpoint: Transplant-free survival	Survival improved by TIPS: 6-weeks: 99 vs. 84%; *p* = 0.02 1-year: 86% vs. 73%; *p* = 0.046 2-years: 79% vs. 64%; *p* = 0.04

ARR: absolute risk reduction; AVB: acute variceal bleeding; CTP: Child–Turcotte–Pugh; EVL: endoscopic variceal ligation; HCC: hepatocellular carcinoma; HE: hepatic encephalopathy; HVPG: hepatic vein pressure gradient; IGV: isolated gastric varices; NSBB: non-selective beta-blockers; PVT: portal vein thrombosis; RCT: randomized clinical trial.

**Table 2 jcm-10-04590-t002:** RCTs assessing TIPS vs. LVP + Albumin in recurrent/refractory ascites.

Reference	Study Population (Randomized Patients)	Exclusion Criteria	Survival Related Endpoints of the Study	Effects on Survival
Lebrec D., et al. (J. Hepatol., 1996)	25 patients with cirrhosis and refractory ascites (no response after 5 days of in-hospital maximal diuretic therapy or ≥2 episodes of tense ascites in the previous 4 months)	Age >70 years, HE ≥grade 2 PVT Biliary obstruction Creatinine >1.7 mg/dL HCC Active bacterial infection Severe extra-hepatic disease Pulmonary hypertension	Not specified	LVP + Albumin vs. TIPS: 2-year overall survival: 60% vs. 29% (*p* = 0.03)
Rossle M., et al. (N. Engl. J. Med., 2000)	60 patients with cirrhosis and refractory ascites or recurrent ascites (ICA criteria)	HE ≥grade 2 PVT Bilirubin >5 mg/dL, Creatinine >3mg/dL Advanced HCC Hepatic hydrothorax Failure of paracentesis(defined as persistence of ascites after paracentesis or need for large-volume paracentesis more than once per week)	Primary endpoint: Transplant-free survival	LVP + Albumin vs. TIPS: 1-year: 69% vs. 58% 2-year: 58% vs. 32% (*p* = 0.11)
Ginès P., et al. (Gastroenterology, 2002)	70 patients with cirrhosis and refractory ascites (ICA criteria)	Age <18/>75 years PVT HE ≥grade 2 Bilirubin >10 mg/dL, Creatinine >3 mg/dL INR >2.5 Platelet <40.000/mm^3^ Chronic Heart Failure HCC Organic renal failure	Primary endpoint: Transplant-free survival	LVP + Albumin vs. TIPS: 1-year: 41% vs. 35% 2-year: 26% vs. 30% (*p* = 0.51)
Sanyal A.J., et al. (Gastroenterology, 2003)	109 patients with cirrhosis and refractory ascites (ICA criteria) plus creatinine <1.5 mg/dL	HE ≥grade 2 PVT Bilirubin >5 mg/dL INR >2 HCC Bacterial infection Alcoholic hepatitis Chronic heart failure Pulmonary hypertension Organic kidney disease Recent gastrointestinal bleeding Severe extra-hepatic disease	Primary endpoint: Overall and transplant-free survival	LVP + Albumin vs. TIPS: Overall: 41.3 vs. 38.2 months; *p* = 0.84 Transplant-free: 19.6 vs. 12.4 months; *p* = 0.77
Salerno F., et al. (Hepatology, 2004)	66 patients with cirrhosis and refractory ascites (ICA criteria) or “recidivant” ascites (recurrence of at least 3 episodes of tense ascites within a 12-month period despite prescription of low sodium diet and adequate diuretic doses)	Age >72 years HE ≥grade 2 PVT CTP score >11 Bilirubin >6 mg/dL Creatinine >3 mg/dL Advanced HCC Bacterial infection Chronic heart failure Recent gastrointestinal bleeding	Primary endpoint: Transplant-free survival	LVP + Albumin vs. TIPS: 1-year: 52% vs. 77% 2-year: 29% vs. 59% (*p* = 0.021)
Narahara Y., et al. (J. Gastroenterol., 2011)	60 patients with cirrhosis and refractory ascites (ICA criteria) plus: CTP score <11 Bilirubin <3 mg/dL Creatinine <1.9 mg/dL	Age >70 years Episodes of HE PV cavernoma HCC Other malignancy Active infection Active severe cardiac or pulmonary disease Organic kidney disease	Primary endpoint: Overall survival	LVP + Albumin vs. TIPS: 1-year: 49% vs. 80% 2-year: 35% vs. 64% (*p* < 0.005)
Bureau C., et al. (Gastroenterology, 2017)	62 patients (>18/<70 year) with cirrhosis and recurrent tense ascites (requiring ≥2 LVP in the previous 3 weeks) PFTE-covered stents	>6 LVPs in the previous 3 months Waitlisted for LT or expected to receive LT within the next 6 months Recurrent overt HE PVT CTP score >12 Bilirubin >5.8 mg/dL Creatinine >2.8 mg/dL HCC Chronic heart failure Pulmonary hypertension	Primary endpoint: Transplant-free survival	LVP + Albumin vs. TIPS: 1-year: 93% vs. 52%; *p* = 0.003

CTP: Child–Turcotte–Pugh; HCC: hepatocellular carcinoma; ICA: International Club of Ascites; INR: international normalized ratio; HE: hepatic encephalopathy; HRS: hepatorenal syndrome; LT: liver transplantation; PFTE: polytetrafluoroethylene; PHT: portal hypertension; PVT: portal vein thrombosis.

## Data Availability

No new data were created or analyzed in this study. Data sharing is not applicable to this article.

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
