# Peer review of "Prevention of Cirrhosis Complications: Looking for Potential Disease Modifying Agents"

_jcm, 2021, doi:10.3390/jcm10194590_

Round 1
Reviewer 1 Report
The manuscript by Zaccherini et al provides an excellent review of the current progress of the efficacy of counteracting key pathophysiological events in preventing cirrhosis complications.
The tables that summarize the results of the available RCTs were especially helpful.
The only minor comment I have is to modify the colors (green/red) in Figure 2 to make it colorblind friendly.
Reviewer 2 Report
This is a review article about therapeutic interventions in decompensated cirrhosis (DC). The manuscript is concise and comprehensive and I mostly agree on the authors' conclusions. In some instances, I may suggest to add some central aspects in DC.
Chapter 3.1, NSBB:
This chapter resembles a compendium about clinical use of NSBB, however it neglects studies investigating the impact of NSBB on portal hypertension. There are valuable clinical studies investigating this issue, which should be included.
3.2 TIPS
- there's a typo at the beginning of the chapter
- The authors should consider to add a paragraph about TIPS in hepatorenal syndrome, since there is a valid treatment rationale, and various prospective trials have been performed on this.
3.3 Antibiotics
Rifaximin is also used for secondary prophylaxis of HE, which should be commented and put into context
3.4 statins
Given that statins have no approval for the treatment of DC whatsoever, this chapter appears too long and prominent in the given context. The pathophysiological rationale and LIVERHOPE study underway must be mentioned, though.
3.5 Albumin
The Attire study must be explizitly mentioned and discussed in this context. Different study results (ANSWER vs. ATTIRE) must be layed out and interpreted accordingly. What is the role of albumin in the management of acute decompensation, compared to long-term care of DC?
3.6 G-CSF
Since G-CSF has recently failed in a large clinical trial investigating ACLF, this chapter appears too prominent, again.
Further comments:
There is no chapter about bacterial infections including multidrug-resistant bacteria. Are there currently any approaches to contain the emergence of MDR?
The authors comment about portal hypertension to a small extent. Moreover, there is no sufficient mentioning of systemic inflammation, which is derived from portal hypertension, and is the main driver of organ dysfunction and organ failures. It would be important if the authors could comment on the results of the DIALIVE study in this context. What about other clinical trials examining plasma exchange in DC and/or ACLF? Are there any other therapeutic approaches directly aiming at systemic inflammation in DC?
In this light, the authors could include a lookout on ACLF prevention and treatment.
Figures:
On a general basis, the figures appear a bit sketchy and should be subjected to a revision of design and layout.
Figure one: How do the authors suggest that 30-50% of DC patients are at risk for death? This number looks like a rough estimate rather than scientifically founded. In any case, according evidence remains unclear within the manuscript. The authors should make an attempt to specify these numbers for every underlying disease, and in case that is not possible omit this otherwise stereotypical figure.
